# Acupuncture for treating attention deficit hyperactivity disorder in children: A protocol for systematic review and meta-analysis

Jung Tae Kim[1,2☯], Kibong Kim[1,3☯], Lin Ang[4], Hye Won Lee[5], Jun-Yong Choi[6], Myeong Soo Lee[4]*

1 Department of Korean Pediatrics, School of Korean Medicine, Pusan National University, Yangsan, Republic of Korea, 2 IMOM Korean Medicine Clinic, Jeju, Republic of Korea, 3 Department of Korean Pediatrics, School of Korean Medicine & Korean Medicine Hospital, Pusan National University, Yangsan, Republic of Korea, 4 KM Science Research Division, Korea Institute of Oriental Medicine, Daejeon, Republic of Korea, 5 KM Convergence Research Division, Korea Institute of Oriental Medicine, Daejeon, Republic of Korea, 6 Department of Korean Internal Medicine, School of Korean Medicine & Korean Medicine Hospital, Pusan National University, Yangsan, Republic of Korea

☯ These authors contributed equally to this work.
* drmslee@gmail.com

**Data Availability Statement:** No datasets were generated or analysed during the current study. All relevant data from this study will be made available upon study completion.

## Abstract

### Background

Attention deficit hyperactivity disorder (ADHD) patients often use complementary and alternative medicine to treat symptoms, and acupuncture is one option. This systematic review aims to assess whether acupuncture is an effective treatment for attention deficit hyperactivity disorder (ADHD).

### Methods

We will search nine databases from their inception: PubMed, AMED, CINAHL, EMBASE, the Cochrane Central Register of Controlled Trials, RISS, KoreaMed, KISS, and the China National Knowledge Infrastructure database. Two investigators will independently review the selected studies, extract the data, and analyze them. The Cochrane Risk of Bias Assessment Tool will be used to assess the risk of bias.

### Discussion

Because this is a systematic review, no ethical approval is needed. The systematic review will be published in a peer-reviewed journal and disseminated both electronically and in print. The review will be updated to support health policy and practice.

### Trial registration number

Reviewregistry1345.

**Funding:** No external funding was supported. LA, HWL and MSL were funded internally by the Korea Institute of Oriental Medicine (KSN2021210) (https://www.kiom.re.kr/eng/). The authors alone are responsible for the writing and content of paper. The funders had and will not have a role in study design, data collection and analysis, decision to publish, or preparation of the manuscript.

**Competing interests:** NO authors have competing interests.

## Introduction

Attention deficit hyperactivity disorder (ADHD) is a disorder with symptoms of inattention, hyperactivity, and impulsivity. The estimated prevalence of ADHD is 7.2% in children and adolescents worldwide [1]. The prevalence of ADHD in children is 7.8%–9.5% in the United States [2] and 6.27% in China [3]. The drug treatment for children with ADHD includes stimulant medication, atomoxetine, and antidepressants in combination with behavioral therapies [4]. However, these therapies are not free of side effects, including abdominal pain, headache, cardiovascular risk, irritability, and insomnia [5].

The reported use of complementary and alternative medicine for ADHD ranges from 12% to 64% [6]. Nutritional interventions, herbal preparations, massage, mind–body medicine, and acupuncture are most commonly used [6]. Acupuncture involves inserting needles for therapeutic or preventive purposes into the skin tissue and underlying tissues at specific points on the body, called acupuncture points [7]. These points can also be stimulated with electricity, lasers, pressure, heat, and ultrasonic waves [7]. Acupuncture is widely used in Asia for managing a variety of conditions, including cardiovascular diseases, infertility, pain and mental health [8–11]. In the cerebral cortex, acupuncture can stimulate the growth and development of nerve fibers, neural nerve regeneration and brain connectivity [12–14]. It has been hypothesized that acupuncture can improve attention, especially active attention, in children with ADHD [15, 16]. To date, the use of acupuncture for ADHD has been investigated in three systematic reviews (SRs) in English [17–19]. One of these three were Cochrane reviews, but they found no eligible studies [19]. The other two reviews indicated limited evidence that acupuncture may be beneficial for the treatment of ADHD [17, 18]. Further, the authors combined the studies regardless of control treatments and ignored clinical heterogeneities, resulting in potentially biased or inaccurate conclusions [17]. They are also outdated and may miss several newly published studies [17, 18]. Therefore, this systematic review will aim to provide and up-to-date evaluation on the effects of acupuncture on children with ADHD.

## Methods

### Study registration

The protocol is registered at reviewregistry1345 (https://www.researchregistry.com/browse-the-registry#registryofsystematicreviewsmeta-analyses/registryofsystematicreviewsmeta-analysesdetails/625ffb6ceccdb1001efe3598/) [20].

### Criteria for considering studies in this review

**Types of studies.** We will include prospective randomized controlled trials (RCTs). Observational, cohort, qualitative, uncontrolled, laboratory, and case-control studies and case series will be excluded. Language restrictions will not apply.

**Types of participants.** Children with ADHD will be included regardless of sex or nationality.

**Types of interventions and controls.** We will include studies that investigated invasive acupuncture with or without electrical stimulation on body, ear and head. Other methods of stimulating acupuncture points without needle insertion (acupressure, pressure buttons, laser stimulation, etc.) will be excluded. Treatments that may be used as control interventions include general conventional treatments (medications), sham treatments (interventions that mimic acupuncture/real treatment in some aspects but differ in others, such as skin penetration or point location), and waiting lists. The acceptance of sham acupuncture as a valid control is highly controversial [21–23], and we will analyze the results using subgroup and

sensitivity analyzes. In addition to acupuncture and another active treatment, we will also include studies that compared the active treatment with acupuncture, as well as studies that combined the active treatment with acupuncture. We will exclude RCTs comparing two different forms of acupuncture or using other types of alternative therapies including cupping, herbal medicines, exercise and etc.

**Type of outcome measures.** *Primary outcomes.* Improvement in ADHD symptoms [24]: DSM-IV-based scales (Conners' Rating Scales (Parent and teacher), Disruptive Behavior Rating Scale, ADHD-RS (Parent, Teacher, and Investigator)) or Global assessment scales (Clinical Global Impression-Improvement or Severity scale, Children's and Parent's Global Assessment Scales).

Secondary outcomes

1. Total treatment efficacy

2. Quality of life

3. Adverse events (AEs)

## Search method for identifying studies

**Electronic searches.** The electronic database searches will be conducted in PubMed, AMED, EMBASE, the Cochrane Central Register of Controlled Trials (CENTRAL), three Korean databases—KoreaMed, the Research Information Service System (RISS), the Korean Studies Information Service System (KISS), and the China National Knowledge Infrastructure (CNKI) database. We will use English for the search terms of all eight databases. In addition, we will use Korean for RISS and KISS and Chinese for CNKI.

Additional studies will be identified by searching the reference lists of the selected studies. We will also search the World Health Organization's International Clinical Trial Registration Platform (ICTRP) (http://apps.who.int/trialsearch), ClinicalTrials.gov (http://clinicaltrials.gov/), and The Clinical Research Information Service (CRIS) (https://cris.nih.go.kr/cris/index/index.do). In addition to the studies located in the database searches, we will include relevant SRs and articles by searching the reference lists of these studies. In addition, dissertations and abstracts will be included in the search.

**Search strategy.** The search strategy will use the following terms: (acupuncture OR acup* OR electroacupuncture OR ear acupuncture) AND (attention deficit OR attention OR hyperactivity* or ADHD).

## Data collection, extraction, and analysis

**Selection of studies.** Two reviewers (JTK, KK) will independently evaluate the titles and abstracts of the studies identified in the searches and select appropriate studies based on the predefined criteria. Disagreements in study selection will be resolved by another reviewer (MSL). A Preferred Reporting Items for Systematic Reviews and Meta-Analysis (PRISMA)-compliant flowchart will be used to document and summarize the study selection [25, 26].

**Data extraction.** The selected articles will be reviewed by two independent reviewers (JTK and LA), who will extract data from the articles based on the predefined criteria. The extracted data will include the acupuncture intervention, the control intervention, the main outcomes, and adverse effects, as well as the author(s) name, year of publication, country, sample size, age, and sex of the study participants. We will tabulate the extracted data for further analysis. GRADE software will be used to assess the strength of evidence based on the

Cochrane Handbook for Systematic Reviews of Interventions, and a table summarizing the results will be produced.

Based on the revised Standards for Reporting Interventions in Clinical Trials of Acupuncture (STRICTA) [27], details of the acupuncture method and control interventions will be extracted. In cases where studies do not meet standards (lack of STRICTA protocol, insufficient information), we will email or call the authors for additional information and note the results in the final publication. We will also report the case where it is not possible to contact the PI of the study and obtain the full information. While all studies that meet the search criteria will be included, those that do not meet the proposed standards will be listed separately and the reason for exclusion is noted. Only those studies that meet the inclusion criteria and none of the exclusion criteria will be included in the results tables.

**Assessment of risk of bias.** We will assess the risk of bias based on the Risk of Bias Assessment Tool (RoB 2.0) developed by the Cochrane Collaboration [28]. Five aspects will be examined: randomization, deviations from planned interventions, missing outcome data, measurement of outcomes, and selection of reported outcomes. Risk of bias will be graded as "low risk of bias," "some concern," or "high risk of bias" for each area of each study. Disagreements will be resolved by involving a third reviewer when necessary.

## Data analysis

The data analysis will be carried out using the Cochrane Collaboration's Review Manager (RevMan) software, version 5.4 for Windows (Nordic Cochrane Center, Copenhagen, Denmark). Comparisons of the intervention and control groups will be performed. The assessment of clinical effectiveness will be based on the risk ratio for the categorical data and the mean difference (MD) for the continuous data. Efficacy values will accompany categorical and continuous variables with 95% confidence intervals. We will use the standardized MD rather than the weighted MD for variables with different scales. When heterogeneity was detected by chi-square or Higgins $I^2$ tests, we will perform subgroup analyses to determine the cause of clinical heterogeneity. Due to the variety of interventions, study designs, and other factors involved in the included studies, a random-effects model will be used to assess the combined effect sizes of the efficacy variables. The Egger regression method and funnel plots will be used to assess publication bias. We will contact the investigators of the original study if we discover missing or incomplete data.

A subgroup analysis will be conducted according to different control interventions (acupuncture vs. sham acupuncture, acupuncture vs. conventional intervention, acupuncture combined with conventional intervention vs. conventional intervention) and stimulation type (manual vs. electrical). The meta-analysis results will be subjected to sensitivity analyses, where appropriate, to determine their robustness.

We will provide all data underlying the results of this study in the article and supplementary materials.

## Ethics and dissemination

The protocol for this systematic review does not require ethical approval. In addition to peer-reviewed publications and conference presentations, the results of this review will be widely disseminated.

## Discussion

Clinicians may be able to utilize the results of this systematic review and meta-analysis on the evidence related to the safety and effectiveness of acupuncture for treating ADHD in children.

## Supporting information

**S1 Checklist. Preferred Reporting Items for Systematic Reviews and Meta-Analyses (PRISMA) checklist.**
(DOC)

**S2 Checklist. STandards for Reporting Interventions in Clinical Trials of Acupuncture (STRICTA) checklist (https://stricta.info/checklist/).**
(PDF)

## Author Contributions

**Conceptualization:** Jung Tae Kim, Kibong Kim, Myeong Soo Lee.

**Data curation:** Lin Ang, Hye Won Lee.

**Formal analysis:** Jung Tae Kim, Lin Ang.

**Investigation:** Jung Tae Kim, Kibong Kim, Myeong Soo Lee.

**Methodology:** Jung Tae Kim, Myeong Soo Lee.

**Project administration:** Hye Won Lee.

**Resources:** Jun-Yong Choi.

**Supervision:** Kibong Kim, Myeong Soo Lee.

**Validation:** Hye Won Lee, Jun-Yong Choi.

**Writing – original draft:** Jung Tae Kim, Kibong Kim, Myeong Soo Lee.

**Writing – review & editing:** Lin Ang, Hye Won Lee, Jun-Yong Choi.

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
