## [Editor Report · Decision Letter 0]

4 Jul 2022

PONE-D-22-17595Acupuncture for treating attention deficit hyperactivity disorder in children: A protocol for systematic review and meta-analysisPLOS ONE

Dear Dr. Lee,

Thank you for submitting your manuscript to PLOS ONE. After careful consideration, we feel that it has merit but does not fully meet PLOS ONE’s publication criteria as it currently stands. Therefore, we invite you to submit a revised version of the manuscript that addresses the points raised during the review process.

We look forward to receiving your revised manuscript.

Kind regards,

Christine Nardini

Academic Editor

PLOS ONE

Journal Requirements:

Additional Editor Comments:

The protocol is very interesting and highly needed, in particular for the absence of language limitations, and for the request of RCT with no constraints on blindeness. However, the presentation of this study requires to be polished, please read below:

“Nowadays, acupuncture is widely

accepted as a treatment method for a variety of conditions, including pain condition and

mental health [8].” This is in fact true with limited geographical application, please rephrase, give context and add citations

“In the cerebral cortex, acupuncture can stimulate the growth and

development of nerve fibers and increase the number and quality of synaptic connections” lacks of reference

“Several studies have hypothesized that acupuncture can improve attention, especially active

attention, in children with ADHD [9].” One citation one study, please rephrase and/or add appropriate citations

“Therefore, the aim of this systematic review and meta-analysis is to provide up-to-date information on the effects of acupuncture on children

with ADHD and to critically evaluate the evidence obtained from randomized trials.” Please state clearly the hypothesis you are testing

“Of these three Cochrane

reviews, one contained no applicable studies [12].” The meaning of this sentence is not clear, what does “applicable studies” mean? Also the authors state that the studies are outdated for this reason only the new work is proposed, discussion is missing as to the limits/outcome of these older meta analyses to justify this proposed work

Acceptance of sham acupuncture as a valid control is highly controversial, and the authors do not discuss this point. Results with and without these studies should probably be provided

“reviewregistry1345” I could not access this study protocol, please add a link or other means to access

2.2.4.1. how will improvement be assessed? Please state criteria

2.3.2 it is not clear why this is separated from 2.3.1, and why not all databases are listed right away.

SR acronym is never explained

Also the authors state that in case of missing (STRICTA) or ambiguous data they will contact the principal investigators, but how will it be possible to recover details on the acupuncture procedure sometimes after years that the study is finished? This should be discussed and clarified and alternatives should also be given (If for instance the PI cannot be reached, or too long a time has passed etc.)

In addition to the PRISMA I would also suggest to add the STRICTA guidelines in the supplementary data
---

## [Author Response · Author response to Decision Letter 0]

5 Jul 2022

Comment 1) When submitting your revision, we need you to address these additional requirements.

Revised> We have now edited the manuscript according to guideline.

The protocol is very interesting and highly needed, in particular for the absence of language limitations, and for the request of RCT with no constraints on blindeness. However, the presentation of this study requires to be polished, please read below:

Comment 2) “Nowadays, acupuncture is widely accepted as a treatment method for a variety of conditions, including pain condition and mental health [8].” This is in fact true with limited geographical application, please rephrase, give context and add citations

Revised> We have now revised the sentence according to comments as follow (page 3, lines 14-16).

Acupuncture is mainly widely used in Asia for managing a variety of conditions, including cardiovascular diseases, infertility, pain and mental health [8-11].

Comment 3) “In the cerebral cortex, acupuncture can stimulate the growth and development of nerve fibers and increase the number and quality of synaptic connections” lacks of reference

Revised> We have now rewritten the sentence and added the references as follow (page 3, lines 7-9 from the bottom).

In the cerebral cortex, acupuncture can stimulate the growth and development of nerve fibers, neural nerve regeneration and brain connectivity [12-14].

Comment 4) “Several studies have hypothesized that acupuncture can improve attention, especially active attention, in children with ADHD [9].” One citation one study, please rephrase and/or add appropriate citations

Revised> We have now rewritten the sentence and added the references as follow (page 3, lines 6-7 from the bottom).

It has been hypothesized that acupuncture can improve attention, especially active attention, in children with ADHD [15, 16].

Comment 5) “Therefore, the aim of this systematic review and meta-analysis is to provide up-to-date information on the effects of acupuncture on children with ADHD and to critically evaluate the evidence obtained from randomized trials.” Please state clearly the hypothesis you are testing.

Revised> We have now rewritten the sentence as suggested (page 4, lines 1-2).

Therefore, this systematic review will aim to evaluate the effects of acupuncture on children with ADHD.

Comment 6) “Of these three Cochrane reviews, one contained no applicable studies [12].” The meaning of this sentence is not clear, what does “applicable studies” mean? 

Revised> We have now changed the words to “eligible studies” (page 3, line 4 from the bottom).

Comment 7) Also the authors state that the studies are outdated for this reason only the new work is proposed, discussion is missing as to the limits/outcome of these older meta analyses to justify this proposed work.

Revised> We have now rewritten the sentence as follows (page 3,lines 1-2 from the bottom and page 4, line 1).

However, the authors combined the studies regardless of control treatments and ignored clinical heterogeneities. They concluded biased or exaggerated conclusions [17]. They are also outdated and may miss several newly published studies [17, 18].

Comment 8) Acceptance of sham acupuncture as a valid control is highly controversial, and the authors do not discuss this point. Results with and without these studies should probably be provided.

Revised> Thank you very much for your valuable. We have now added the sentences with references as follows (page 5, lines 2-4).

The acceptance of sham acupuncture as a valid control is highly controversial [21-23], and we will analyze the results using subgroup and sensitivity analyzes.

Comment 9) “reviewregistry1345” I could not access this study protocol, please add a link or other means to access

Revised> There was a link as reference [20] in the original submission. We have now added the link in the text (page 4, lines 6-7).

(https://www.researchregistry.com/browse-the-registry#registryofsystematicreviewsmeta-analyses/registryofsystematicreviewsmeta-analysesdetails/625ffb6ceccdb1001efe3598/)

Comment 10) 2.2.4.1. how will improvement be assessed? Please state criteria.

Revised> We have now added the core-outcomes related to measuring improvement (core-outcomes) with references as follow (lines 11-14).

Improvement in ADHD symptoms [24]: DSM-IV-based scales (Conners’ Rating Scales (Parent and teacher), Disruptive Behavior Rating Scale, ADHD-RS (Parent, Teacher, and Investigator)) or Global assessment scales (Clinical Global Impression-Improvement or Severity scale, Children’s and Parent’s Global Assessment Scales)

Comment 11) 2.3.2 it is not clear why this is separated from 2.3.1, and why not all databases are listed right away.

Revised> We have now combined two sections as one and added the details as follow (page 5, lines 6-7).

The Clinical Research Information Service (CRIS) (https://cris.nih.go.kr/cris/index/index.do).

Comment 12) SR acronym is never explained.

Reply> It is already spelled out in the page 3, line 4 from the bottom.

Comment 13) Also the authors state that in case of missing (STRICTA) or ambiguous data they will contact the principal investigators, but how will it be possible to recover details on the acupuncture procedure sometimes after years that the study is finished? This should be discussed and clarified and alternatives should also be given (If for instance the PI cannot be reached, or too long a time has passed etc.)

Revised> Thank you for your comments. We have now added the sentence for the commented cases (page 7, lines 8-9). 

If the principal investigators cannot be reached or too much time has elapsed, we will note this in the full review reports. 

Comment 14) In addition to the PRISMA I would also suggest to add the STRICTA guidelines in the supplementary data

Reply> We have now added it as supplementary file.

---

## [Decision Letter · Decision Letter 1]

2 Aug 2022

PONE-D-22-17595R1Acupuncture for treating attention deficit hyperactivity disorder in children: A protocol for systematic review and meta-analysisPLOS ONE

Dear Dr. Lee,

Thank you for submitting your manuscript to PLOS ONE. After careful consideration, we feel that it has merit but does not fully meet PLOS ONE’s publication criteria as it currently stands. Therefore, we invite you to submit a revised version of the manuscript that addresses the points raised during the review process.

We look forward to receiving your revised manuscript.

Kind regards,

Christine Nardini

Academic Editor

PLOS ONE

Additional Editor Comments:

It is essential that you clarify how you will handle cases where studies do not comply to standards (lack of STRICTA protocol, lack of sufficient information, impossibility to contact the PI of the study, etc) and in which cases you will discard the study for lack of quality

Reviewers' comments:

Reviewer's Responses to Questions

**Comments to the Author**

1. Does the manuscript provide a valid rationale for the proposed study, with clearly identified and justified research questions?

Reviewer #1: Yes

2. Is the protocol technically sound and planned in a manner that will lead to a meaningful outcome and allow testing the stated hypotheses?

Reviewer #1: Yes

3. Is the methodology feasible and described in sufficient detail to allow the work to be replicable?

Reviewer #1: Yes

4. Have the authors described where all data underlying the findings will be made available when the study is complete?

Reviewer #1: No

5. Is the manuscript presented in an intelligible fashion and written in standard English?

Reviewer #1: Yes

6. Review Comments to the Author

You may also provide optional suggestions and comments to authors that they might find helpful in planning their study.

Reviewer #1: Page 3 - "Acupuncture is mainly widely used" - 'mainly widely' does not read well. Suggest rephase to "Acupuncture is widely used in Asia"

Page 3 - five lines from the bottom "one contained no eligible studies". I do not understand what the authors mean here. If no studies were eligible, does it mean that the review showed that there are zero studies that asses the use of acupuncture in children with ADHD? Eligible in what sense?

Page 3 - last line "They concluded biased or exaggerated conclusions". This almost seems like a contradiction to the previous comment that these studies indicted limited evidence. If the evidence is limited and exaggerated, this implies that there is in fact no evidence. I would suggest "Further, the authors combined the studies regardless of control treatments and ignored clinical heterogeneities, resulting in potentially biased or inaccurate conclusions"

Page 4 - line 2. Rephase "...systematic review will aim to provide and up-to-date evaluation on the effects...."

Page 4 - Under types of studies. Specify which languages will be used for the searches.

Page 4 to 5 - In types of interventions and controls, how will studies that include alternative therapies alongside acupuncture (eg cupping, exercise) be handled?

Page 5 lines 5 to 6, I find this sentence clumsy. Rephrase the sentence to clarify that combined treatments will be considered eg. "we will also include studies that compared the active treatment with acupuncture, as well as studies that combined the active treatment with acupuncture"

Page 6 - "searching reference lists of the selected studies" is written 3 times on this page.

Page 6 search strategy. Is it worth adding "ADHD" to the search?

Page 7 Assessment of risk bias. This paragraph has been written in the past tense. I think it should be written in future tense, assuming it is relevant to the proposed analyses.

Data: Given that the authors propose performing subgroup analyses, data from studies included in the review will need to be obtained / downloaded. These data should be made available and how this will be done should be discussed within the text.

7. PLOS authors have the option to publish the peer review history of their article (what does this mean?). If published, this will include your full peer review and any attached files.

Reviewer #1: No

---

## [Author Response · Author response to Decision Letter 1]

2 Aug 2022

Dear Editor

On the behalf of my co-author, I would like to thank you for arranging peer-review of our manuscript and for your invitation to submit a revised version. We have highlighted the revised points according to valuable comments. We appreciate the effort of the reviewers and believe that their constructive suggestions have resulted in a stronger manuscript for the PLoS One’s readers.

Yours faithfully,

Myeong Soo Lee, PhD on the behalf of co author

Additional Editor Comments:

Comment 1) It is essential that you clarify how you will handle cases where studies do not comply to standards (lack of STRICTA protocol, lack of sufficient information, impossibility to contact the PI of the study, etc) and in which cases you will discard the study for lack of quality

Revised> Thank you very much for your valuable comments. We have now changed the original sentences as following sentences (P. 7, Data extraction, 2nd paragraph).

“In cases where studies do not meet standards (lack of STRICTA protocol, insufficient information), we will email or call the authors for additional information and note the results in the final publication. We will also report the case where it is not possible to contact the PI of the study and obtain the full information. We will include all eligible studies, regardless of missing information and poor quality, to obtain complete evidence.”

Reviewers' comments:

Comment 1): Page 3 - "Acupuncture is mainly widely used" - 'mainly widely' does not read well. Suggest rephase to "Acupuncture is widely used in Asia"

Revised> We have now changed it as commented (P. 3, 2nd paragraph, line 6).

Comment 2) Page 3 - five lines from the bottom "one contained no eligible studies". I do not understand what the authors mean here. If no studies were eligible, does it mean that the review showed that there are zero studies that asses the use of acupuncture in children with ADHD? Eligible in what sense?

Revised> Thank you so much for your valuable comment. We have now clarified this as followings (P.3, lines 4-5 from the bottoms).

“One of these three were Cochrane reviews, but they found no eligible studies [19].”

Comment 3) Page 3 - last line "They concluded biased or exaggerated conclusions". This almost seems like a contradiction to the previous comment that these studies indicted limited evidence. If the evidence is limited and exaggerated, this implies that there is in fact no evidence. I would suggest "Further, the authors combined the studies regardless of control treatments and ignored clinical heterogeneities, resulting in potentially biased or inaccurate conclusions"

Revised> We have now changed it according to your valuable comments (P 3, lines 1-2 from the bottoms).

Comment 4) Page 4 - line 2. Rephase "...systematic review will aim to provide and up-to-date evaluation on the effects...."

Revised> We have now changed it according to your valuable comments (P. 4, line 2).

Comment 5) Page 4 - Under types of studies. Specify which languages will be used for the searches.

Revised> Thank you so much for your valuable comment. We have now added this under the Electric searches as followings (P. 6, lines 5-6).

“We will use English for the search terms of all eight databases. In addition, we will use Korean for RISS and KISS and Chinese for CNKI.”

Comment 6) Page 4 to 5 - In types of interventions and controls, how will studies that include alternative therapies alongside acupuncture (eg cupping, exercise) be handled?

Revised> Thank you so much for your valuable comment. We have now added this as followings (P. 5, lines 8-9).

“We will exclude RCTs comparing two different forms of acupuncture or using other types of alternative therapies including cupping, herbal medicines, exercise and etc.”

Comment 7) Page 5 lines 5 to 6, I find this sentence clumsy. Rephrase the sentence to clarify that combined treatments will be considered eg. "we will also include studies that compared the active treatment with acupuncture, as well as studies that combined the active treatment with acupuncture"

Revised> We have now changed it according to your suggestion (P. 5, lines 6-8).

Comment 8) Page 6 - "searching reference lists of the selected studies" is written 3 times on this page.

Revised> We have now corrected them according to your valuable comments (P. 6, 2nd paragraph, lines 5-7).

Comment 9) Page 6 search strategy. Is it worth adding "ADHD" to the search?

Revised> We have now added it according to your valuable comments (P. 6, Search strategies, line 3).

Comment 10) Page 7 Assessment of risk bias. This paragraph has been written in the past tense. I think it should be written in future tense, assuming it is relevant to the proposed analyses.

Revised> We have now corrected them (P. 7, lines 2-3 from the bottoms, P. 8, lines 1-2). 

"We will assess the risk of bias based on the Risk of Bias Assessment Tool (RoB 2.0) developed by the Cochrane Collaboration [28]. Five aspects will be examined: randomization, deviations from planned interventions, missing outcome data, measurement of outcomes, and selection of reported outcomes. Risk of bias will be graded as "low risk of bias," "some concern," or "high risk of bias" for each area of each study. Disagreements will be resolved by involving a third reviewer when necessary."

Comment 11) Data: Given that the authors propose performing subgroup analyses, data from studies included in the review will need to be obtained / downloaded. These data should be made available and how this will be done should be discussed within the text.

-----------------

Have the authors described where all data underlying the findings will be made available when the study is complete?

-------------

Revised> We have now added it as followings according to your valuable comments (P. 9, lines 1-2. P.10. Data availability).

P.9, Lines 1-2

“We will provide all data underlying the results of this study in the article and supplementary materials.”

Data availability

“The authors confirm that the data supporting the findings of this study will be fully available within the article and its supplementary materials. Further inquiries can be directed to the corresponding author.”

---

## [Decision Letter · Decision Letter 2]

8 Sep 2022

PONE-D-22-17595R2Acupuncture for treating attention deficit hyperactivity disorder in children: A protocol for systematic review and meta-analysisPLOS ONE

Dear Dr. Lee,

Thank you for submitting your manuscript to PLOS ONE. After careful consideration, we feel that it has merit but does not fully meet PLOS ONE’s publication criteria as it currently stands. Therefore, we invite you to submit a revised version of the manuscript that addresses the points raised during the review process.

We look forward to receiving your revised manuscript.

Kind regards,

Christine Nardini

Academic Editor

PLOS ONE

Additional Editor Comments:

Dear Authors,

as I noted earlier I again highlight the necessity to be crystal clear on the inclusion and exclusion criteria for the studies of the meta analysis.

This is the only, but crucial point that needs to be clearly addressed before the article can be published.

Reviewers' comments:

Reviewer's Responses to Questions

**Comments to the Author**

1. Does the manuscript provide a valid rationale for the proposed study, with clearly identified and justified research questions?

Reviewer #1: Yes

2. Is the protocol technically sound and planned in a manner that will lead to a meaningful outcome and allow testing the stated hypotheses?

Reviewer #1: Yes

3. Is the methodology feasible and described in sufficient detail to allow the work to be replicable?

Reviewer #1: Yes

4. Have the authors described where all data underlying the findings will be made available when the study is complete?

Reviewer #1: Yes

5. Is the manuscript presented in an intelligible fashion and written in standard English?

Reviewer #1: Yes

6. Review Comments to the Author

You may also provide optional suggestions and comments to authors that they might find helpful in planning their study.

Reviewer #1: Thank you for taking the time to make the proposed changes. I am satisfied that all previous comments have been considered. I have only one remaining concern regarding the following comment:

"We will include all eligible studies, regardless of missing information and poor quality, to obtain complete evidence."

I do not agree that poor quality studies should be included as this may lead to misuse and poor-quality results further down the line. If the authors wish to include these studies in their report, then I think that it should be explicitly stated that "whilst all studies that meet the search criteria will be considered, those that do not meet the proposed standards will be listed separately with the reason for exclusion noted". Reasons for exclusion should then be listed in a manner that it not ambiguous, eg "insufficient information" should state what was missing and how much was missing. Only those studies that meet the inclusion criteria and none of the exclusion criteria should be presented in the results tables.

7. PLOS authors have the option to publish the peer review history of their article (what does this mean?). If published, this will include your full peer review and any attached files.

Reviewer #1: No

---

## [Author Response · Author response to Decision Letter 2]

8 Sep 2022

Dear Reviewer,

On the behalf of my co-author, I would like to thank you for arranging peer-review of our manuscript and for your invitation to submit a revised version. We have highlighted the revised points according to valuable comments. We appreciate the effort of the reviewers and believe that their constructive suggestions have resulted in a stronger manuscript for the PLoS One’s readers.

Yours faithfully,

Myeong Soo Lee, PhD on the behalf of co author

Reviewers' comments:

Comment 1) Thank you for taking the time to make the proposed changes. I am satisfied that all previous comments have been considered. I have only one remaining concern regarding the following comment:

"We will include all eligible studies, regardless of missing information and poor quality, to obtain complete evidence."

I do not agree that poor quality studies should be included as this may lead to misuse and poor-quality results further down the line. If the authors wish to include these studies in their report, then I think that it should be explicitly stated that "whilst all studies that meet the search criteria will be considered, those that do not meet the proposed standards will be listed separately with the reason for exclusion noted". Reasons for exclusion should then be listed in a manner that it not ambiguous, eg "insufficient information" should state what was missing and how much was missing. Only those studies that meet the inclusion criteria and none of the exclusion criteria should be presented in the results tables.

Revised> We have now changed it as commented (P. 7, lines 2-5 from the bottoms).

“While all studies that meet the search criteria will be included, those that do not meet the proposed standards will be listed separately and the reason for exclusion is noted. Only those studies that meet the inclusion criteria and none of the exclusion criteria will be included in the results tables.”

---

## [Editor Report · Decision Letter 3]

19 Sep 2022

Acupuncture for treating attention deficit hyperactivity disorder in children: A protocol for systematic review and meta-analysis

PONE-D-22-17595R3

Dear Dr. Lee,

We’re pleased to inform you that your manuscript has been judged scientifically suitable for publication and will be formally accepted for publication once it meets all outstanding technical requirements.

Kind regards,

Christine Nardini

Academic Editor

PLOS ONE
---

## [Editor Report · Acceptance letter]

30 Sep 2022

PONE-D-22-17595R3 

Acupuncture for treating attention deficit hyperactivity disorder in children: A protocol for systematic review and meta-analysis 

Dear Dr. Lee:

I'm pleased to inform you that your manuscript has been deemed suitable for publication in PLOS ONE. Congratulations! Your manuscript is now with our production department. 

Kind regards, 

on behalf of

Dr. Christine Nardini 

Academic Editor

PLOS ONE